# E3 Ubiquitin Ligase FBXO3 Drives Neuroinflammation to Aggravate Cerebral Ischemia/Reperfusion Injury

**DOI:** 10.3390/ijms232113648

**Published:** 2022-11-07

**Authors:** Yu Gao, Xinyu Xiao, Jing Luo, Jianwei Wang, Qiling Peng, Jing Zhao, Ning Jiang, Yong Zhao

**Affiliations:** 1Department of Pathology, Chongqing Medical University, Chongqing 400016, China; 2Molecular Medicine Diagnostic and Testing Center, Chongqing Medical University, Chongqing 400016, China; 3School of Basic Medical Science, Chongqing Medical University, Chongqing 400016, China; 4Institute of Neuroscience, Chongqing Medical University, Chongqing 400016, China; 5Department of Pathophysiology, Chongqing Medical University, Chongqing 400016, China

**Keywords:** ischemic stroke, cerebral ischemia/reperfusion injury, inflammation, neuronal damage, FBXO3, HIPK2

## Abstract

Ischemic stroke, one of the most universal causes of human mortality and morbidity, is pathologically characterized by inflammatory cascade, especially during the progression of ischemia/reperfusion (I/R) injury. F-Box Protein 3 (FBXO3), a substrate-recognition subunit of SKP1-cullin 1-F-box protein (SCF) E3 ligase complexes, has recently been proven to be severed as an underlying pro-inflammatory factor in pathological processes of diverse diseases. Given these considerations, the current study aims at investigating whether FBXO3 exerts impacts on inflammation in cerebral I/R injury. In this study, first, it was verified that FBXO3 protein expression increased after a middle cerebral artery occlusion/reperfusion (MCAO/R) model in Sprague–Dawley (SD) rats and was specifically expressed in neurons other than microglia or astrocytes. Meanwhile, in mouse hippocampal neuronal cell line HT22 cells, the elevation of FBXO3 protein was observed after oxygen and glucose deprivation/reoxygenation (OGD/R) treatment. It was also found that interference of FBXO3 with siRNA significantly alleviated neuronal damage via inhibiting the inflammatory response in I/R injury both in vivo and in vitro. The FBXO3 inhibitor BC-1215 was used to confirm the pro-inflammatory effect of FBXO3 in the OGD/R model as well. Furthermore, by administration of FBXO3 siRNA and BC-1215, FBXO3 was verified to reduce the protein level of Homeodomain-Interacting Protein Kinase 2 (HIPK2), likely through the ubiquitin–proteasome system (UPS), to aggravate cerebral I/R injury. Collectively, our results underline the detrimental effect FBXO3 has on cerebral I/R injury by accelerating inflammatory response, possibly through ubiquitylating and degrading HIPK2. Despite the specific interaction between FBXO3 and HIPK2 requiring further study, we believe that our data suggest the therapeutic relevance of FBXO3 to ischemic stroke and provide a new perspective on the mechanism of I/R injury.

## 1. Introduction

Ischemic stroke, one of the most universal causes of human mortality and morbidity [1] with up to 12 million cases in 2019 worldwide [2], is induced by an immediate shutdown of brain blood flow. Despite the recanalization of the occluded cerebral artery, neuronal damage continues due to sudden reperfusion of blood and oxygen in ischemic parenchyma, which is called cerebral ischemia/reperfusion (I/R) injury. Increasing evidence has illustrated that inflammation is an essential pathological factor in I/R injury [3,4]. When cells in ischemic penumbra suffering from I/R injury release danger-associated molecular patterns (DAMPs) [5], immune cells are recruited, produce a series of pro-inflammatory cytokines to magnify the inflammatory cascade, and exacerbate initial neuronal damage [6]. Effective therapy for avoiding or easing the damage that the inflammatory cascade causes is urgently needed. However, the detailed mechanisms of inflammation regulating cerebral I/R injury have not been elucidated clearly.

FBXO3, F-Box Protein 3, is the substrate-recognition subunit of SKP1-cullin 1-F-box protein (SCF) E3 ligase complexes [7]. The most important part of the ubiquitin–proteasome system (UPS) [8], E3 is involved in protein degradation throughout the human body, as well as pathological processes including disorders of the central nerve system (CNS) [9]. Mutations of FBXO7 are considered as pathogenic variants of parkinsonian-pyramidal syndrome [10], with mechanisms including neurodegeneration, mitochondrial damage, oxidative stress, and mitophagy [11]. Schneider et al. [12] reported that every FBXO28 microdeletion case suffers from epilepsy, and most also have mental retardation, dyskinesia, and microcephaly. Interestingly, FBXO3 gene expression has been found to be highly correlated to epilepsy using high-throughput gene expression analysis [13]. Beyond the connection with CNS, FBXO3 has been demonstrated to mediate ubiquitination degradation of F-Box and Leucine Rich Repeat Protein 2 (FBXL2) [14], thereby promoting inflammatory cytokines’ expression [15] and rescuing the expression of NLR Family Pyrin Domain Containing 3 (NLRP3) [16], which has been confirmed to prompt inflammatory cascade and neuronal damage in ischemic stroke [17,18]. Aside from the connection between FBXO3 and inflammation mentioned above, FBXO3 has been reported to promote inflammation in several kinds of diseases, including pneumonia, sepsis [14], and atherosclerosis [19]. In addition, FBXO3 inhibitor BC-1215 attenuates ischemia–reperfusion-induced lung injury in rat models [20]. Yet, whether FBXO3 regulates inflammation in cerebral I/R injury is fully uninvestigated.

A previous study has revealed Homeodomain-Interacting Protein Kinase 2 (HIPK2), a nuclear serine/threonine kinase, as a convincing target for FBXO3-mediated ubiquitination [21], and the bioinformatics database also displays FBXO3 interacting with and ubiquitinating HIPK2 [22]. As a member of the evolutionarily conserved HIPK family [23], HIPK2 is characterized as a corepressor or a coactivator, through binding and phosphorylating target proteins [24,25]. It is demonstrated that overexpression of HIPK2 attenuates inflammation, leading to the remission of spinal cord injury in rats [26], which implies a connection between HIPK2 and inflammation. Furthermore, HIPK2 overexpression mitigates myocardial hypoxia/reoxygenation injury by improving the nuclear expression and transcriptional activity of Nuclear factor (erythroid-derived 2)-like 2 (Nrf2) [27]. Moreover, HIPK2 and Nrf2 seem to have crosstalk: not only does Nrf2 stabilize the levels of HIPK2, but HIPK2 also promotes the antioxidant response induced by Nrf2 [28]. Given our previous study that Nrf2 inhibits the inflammatory cascade caused by the NLRP3 inflammasome, thereby alleviating cerebral I/R injury in Sprague–Dawley (SD) rats [29], the effect of HIPK2 on inflammation in ischemic stroke is worthy of inquiry. Additionally, the explicit function of FBXO3 and its downstream target HIPK2 in I/R injury remains obscure.

In the present study, the effects of FBXO3 are preliminarily explored that the levels of FBXO3 are upregulated in an I/R model, and the suppression of FBXO3 elevates neuronal survival both in vivo and in vitro. Additionally, inhibition of FBXO3 alleviates inflammatory response to protect against I/R injury. Furthermore, our results provide initial evidence that FBXO3 aggravates inflammation, probably by degrading HIPK2 in an oxygen and glucose deprivation/reoxygenation (OGD/R) model. Taken together, our data suggest the therapeutic relevance of FBXO3 to ischemic stroke and provide a new perspective on the mechanism of I/R injury.

## 2. Results

### 2.1. FBXO3 Was Significantly Elevated in the Peri-Infarcted Brain Tissue of SD Rats Subjected to Middle Cerebral Artery Occlusion/Reperfusion (MCAO/R) and Specifically Expressed in Neurons

To investigate the involvement of FBXO3 in the progression of ischemic stroke, MCAO was performed, and samples were harvested at different times of reperfusion. Western blot (WB) results showed that, compared to the Sham group, FBXO3 protein expression was increased after 6 h of reperfusion and peaked at 24 h post-stroke (Sham group vs. MCAO 1 h/R 24 h group, *p* < 0.01) (Figure 1A,B), so these conditions were chosen for subsequent experiments. Immunofluorescence results from ischemic penumbra exhibited abundant FBXO3 expression in neurons (NeuN, neuronal biomarker) [30] (Figure 1C), which was negligible in microglia (Iba-1, microglial biomarker) [31] (Figure 1D) and astrocytes (GFAP, astrocytic biomarker) [32] (Figure 1E), suggesting its specific expression in the neurons of the cortex. Additionally, there were more FBXO3+ neurons in the MCAO group than in the Sham group (Figure 1C). The above results indicated that the protein level of FBXO3 was elevated after MCAO, and specifically expressed in neurons.

### 2.2. Interference of FBXO3 Rescued the Neurological Outcomes after MCAO/R

To assess the effect of FBXO3 in I/R injury, si-FBXO3 was injected in the left lateral cerebral ventricle of SD rats prior to MCAO/R. Immunoblot and qPCR results identified the successful transfection of si-FBXO3 (Figure 2A–C and Appendix A). TTC (2,3,5-triphenyltetrazolium chloride) staining was applied to visualize cerebral infarcted areas, as shown in Figure 2D,E: infarcted areas appeared white, while nonimpaired areas were red, revealing that the infarct volumes of MCAO group were significantly observed, and partially recovered by si-FBXO3 (NC group vs. si-FBXO3 group, *p* < 0.05). Likewise, neurological deficit scores were representative of damage: the higher the scores, the more damage rats have in their brain. The Sham group scored 0, the MCAO group had the highest scores, and the si-FBXO3 group lowered the scores (NC group vs. si-FBXO3 group, *p* < 0.05) (Figure 2F), implying that the inhibition of FBXO3 could improve the neurological impairments. HE staining was applied to display the histological and morphological alterations of infarct brains, as shown in Figure 2G. The cerebral cortex in peri-infarcted areas was characterized as disordered organization with edema, loosened cytoplasm, and cell necrosis including pyknosis, karyorrhexis, and karyolysis in the MCAO and NC groups, whereas si-FBXO3 interference rescued the tissue damage and limited the number of dead neurons. Furthermore, Nissl staining exhibited an apparent increase in Nissl bodies in the si-FBXO3 group compared to the NC group (Figure 2G), implying that interference with FBXO3 improves neuronal survival. Collectively, these results suggest that FBXO3 aggravates I/R outcomes after stroke.

### 2.3. Interference of FBXO3 Attenuated OGD/R-Induced Neuronal Death

Having identified the specific expression of FBXO3 in neurons in vivo, we selected neuronal cell line HT22 cells to exemplify how FBXO3 influences I/R injury in vitro. The FBXO3 level was found to increase after OGD 4 h/R 3 h and peak at 24 h of reoxygenation (Control group vs. OGD 4 h/R 24 h group, *p* < 0.05), as exhibited in the WB results (Figure 3A,C), so these conditions were chosen for the following experiments. Consistent with the in vivo results, the immunofluorescence results showed that, in HT22 cells, compared to the Control group, the fluorescence intensity of FBXO3 was higher in OGD/R group, and colocalization of NeuN and FBXO3 was observed (Appendix A). Following verification of the alteration of FBXO3 expression in HT22 cells, siRNA of FBXO3 was applied to successfully transfect neurons, as shown in Figure 3B,D,E and Appendix A. When we examined the survival rate of HT22 cells in the OGD/R model, the CCK8 results showed that the survival rate, which was lowered by OGD/R treatment (Control group vs. OGD group, *p* < 0.01), was elevated by si-FBXO3 transfection (NC group vs. si-FBXO3 group, *p* < 0.001) (Figure 3F), and the flowcytometry results also demonstrated that si-FBXO3 apparently attenuated the neuronal apoptosis caused by OGD/R exposure (NC group vs. si-FBXO3 group, *p* < 0.001) (Figure 3G,H). A Calcein/PI Cell Viability/Cytotoxicity assay was performed to strengthen the reliability of our results, showing that si-FBXO3 apparently reduced the number of dead neurons and improved the morphology of HT22 cells (Figure 3I). In summary, the above in vitro results extended our theory that the level of FBXO3 was increased after ischemic stroke, and specifically expressed in neurons, illustrating that FBXO3 may directly regulate neuronal function in I/R injury.

### 2.4. Treatment of si-FBXO3 Inhibited Inflammatory Response after I/R Injury In Vivo and In Vitro

Given that FBXO3 plays roles in inflammation in other disease models [14], this study was designed to assess whether FBXO3 mediated the inflammation induced by I/R injury. First, in vivo, WB results indicated that the protein levels of proinflammatory cytokines IL-1β, IL-18, and TNFα were significantly enhanced in the MCAO group (MCAO group vs. Sham group, *p* < 0.01 in every cytokine type), whereas FBXO3 siRNA transfection rescued the upregulation induced by MCAO/R (NC group vs. si-FBXO3 group, *p* < 0.05 for IL-1β and TNFα, *p* < 0.01 for IL-18) (Figure 4A,B), with which subsequent ELISA results were consistent (Figure 4C,D), indicating that FBXO3 may stimulate inflammation during the progression of I/R injury in SD rats. Similar to the results above, the in vitro results illustrated that FBXO3 interference profoundly attenuated the expression of proinflammatory cytokines induced by OGD/R (Figure 5A–D). Moreover, myeloperoxidase (MPO), widely known as a critical indicator of inflammation [33], was detected in the present study. MPO-positive cells were significantly decreased in the si-FBXO3 group compared to the NC group (Figure 4E). To further confirm the reliability of our results, we utilized FBXO3 inhibitor BC-1215 to identify the effect of FBXO3 on inflammation in the OGD/R model. Our results indicated that BC-1215 involvement could downregulate the level of inflammatory cytokines (Figure 5E,F), which was supported by the corresponding ELISA results (Figure 5G,H). Based on these experiments, we reason that FBXO3 could mediate the inflammatory response of neurons to exacerbate neuronal damage in I/R injury.

### 2.5. FBXO3 Facilitated Inflammation, Probably through Binding and Degrading HIPK2 in HT22 Cells after OGD/R Stimulation

HIPK2, as a convincing anti-inflammatory cytokine and a ubiquitination target of FBXO3 [21], was explored in our study for its involvement in FBXO3-mediated cerebral I/R injury in vitro. First, it was shown that HIPK2 protein expression declined after OGD/R compared with the Control group (*p* < 0.05), whose change was reversed by FBXO3 interference (NC group vs. si-FBXO3 group, *p* < 0.05) (Figure 6A,C), indicating that FBXO3 could negatively regulate HIPK2 expression in HT22 cells suffering from OGD/R. It is worth noting that BC-1215 targets the ApaG domain, the substrate recognition domain of FBXO3, to prevent FBXO3 from binding with substrates and causing their degradation [15]. As shown in Figure 6B,D and Appendix A, our results also demonstrated that BC-1215 upregulated the protein expression levels of HIPK2 in the OGD/R model of HT22 cells (BC-1215 group vs. DMSO group, *p* < 0.01), but had no effect on FBXO3 levels. Furthermore, the immunofluorescence results indicated that FBXO3 was colocalized with HIPK2 in the nucleus and cytoplasm of HT22 cells, and the fluorescence intensity of HIPK2 was negatively regulated by FBXO3 (Figure 6F). In addition, Co-IP results verified the interaction of FBXO3 and HIPK2 (Figure 6E). Given all the results mentioned, it is reasonable to infer that FBXO3 may ubiquitylate and degrade HIPK2 to mediate inflammation, thereby aggravating cerebral I/R injury.

## 3. Discussion

In our study, FBXO3 was crucial for the progression of cerebral I/R injury. The FBXO3 level was elevated in neurons after stroke, and siRNA-induced FBXO3 knockdown rescued the neurological deficits caused by I/R injury in SD rats, while si-FBXO3 was also demonstrated to revise neuronal survival in vitro. In addition, we found that si-FBXO3 significantly inhibited inflammation in both in vivo and in vitro experiments. Furthermore, inhibition of FBXO3 by siRNA or BC-1215 not only suppressed the levels of inflammatory cytokines but also increased HIPK2 expression in the OGD/R model. More importantly, our results showed that FBXO3 and HIPK2 were co-expressed and interacted in neurons. Previous evidence suggests that HIPK2 may serve as a ubiquitinated substrate of FBXO3. Collectively, it is reasonable to assume that FBXO3 may ubiquitylate and degrade HIPK2 to mediate inflammation, thereby aggravating cerebral I/R injury (Figure 7).

FBXO3 is one member of the F-box family, identified as a component of ubiquitin ligase E3 to participate in UPS [7]. As known, UPS governs more than 80% of degradations of cellular proteins [34], in which E3 ligase plays a decisive role: it specifically recognizes substrates and promotes the translocation of ubiquitin to substrates [35]. Known that SCF E3 complex is the largest E3 family, it is widely involved in physiological and pathological processes. Since being discovered [36], FBXO1 has been confirmed to be closely connected to cancer in that FBXO1 inhibits tumorigenesis of gliomas owing to interacting with the recombination signal binding protein for immunoglobulin kappa J region (RBPJ) for degradation [37]. In addition, Suber et al. [38] discovered that FBXO17 constituting the SCF^FBXO17^ E3 ubiquitin ligase complex regulates inflammation by ubiquitinating and degrading GSK3β in lung epithelia. Interestingly, FBXO3 promotes the ubiquitylation and degradation of ΔNp63α to enhance TGF-β signaling and accelerate tumor metastasis, indicating that FBXO3 may be a potential therapeutic target for advanced breast cancer treatment [39]. In neuropathic allodynia, FBXO3 serves as a deleterious factor through the TRAF2/TNIK/GluR1 cascade [40], indicating a connection between FBXO3 and neurons. Here, our results demonstrated that FBXO3 protein expression was increased after I/R injury, and specifically expressed in neurons; furthermore, in an SD rat MCAO/R model, si-FBXO3 could partially recover the infarct volumes and lower the neurological deficit scores, while in HT22 cells, si-FBXO3 transfection elevated the neuronal survival rate and attenuated the neuronal apoptosis caused by OGD/R. Moreover, FBXO3 is known to stimulate cytokine secretion by destabilizing FBXL2 through UPS [14], and exert its pro-inflammatory effect in a variety of disease models including lung I/R injury [20], pneumonia, sepsis [14], and atherosclerosis [19]. In concert with these findings, our results revealed that si-FBXO3 decreased the expression of IL-1β, IL-18, and TNF-α, verifying that FBXO3 is a pro-inflammatory regulator in I/R injury both in vivo and in vitro. Furthermore, to increase the credibility of our conclusions, BC-1215, an inhibitor of FBXO3, was administered to verify that it could downregulate the levels of inflammatory cytokines caused by OGD/R in HT22 cells, coinciding with the results of Weathington et al. [41], in which BC1215 restrains NF-κB activity in acute respiratory distress syndrome (ARDS). Taken together, FBXO3 exacerbated cerebral I/R injury by aggravating inflammation.

Given that HIPK2 is a convincing ubiquitination target of FBXO3 [21], and that the bioinformatics database also shows FBXO3 interacting and ubiquitinating HIPK2 [22], a reasonable conclusion is that FBXO3 mediates inflammation by ubiquitinating HIPK2. HIPK2, a nuclear protein kinase, is known to participate in various biological processes, such as cell proliferation, apoptosis, and DNA damage response [42]. The dysregulation of HIPK2 leads to diabetes [43], myocardial infarction [44], and colitis-associated diseases, including colorectal carcinoma and sepsis [25]. In addition, overexpression of HIPK2 alleviates LPS-induced inflammation in macrophages, and HIPK2-KO mice are more susceptible to LPS-stimulated endotoxemia [25]. In spinal cord injury (SCI), HIPK2 overexpression eases pain by several mechanisms that include inflammatory response [26]. The above studies imply that HIPK2 is an anti-inflammation factor. In the present study, HIPK2 protein expression was reduced after OGD/R relative to the Control group, while si-FBXO3 or FBXO3 inhibitor BC-1215 could rescue its protein expression. What is more, FBXO3 was shown to colocalize and bind with HIPK2 in HT22 cells. It is noteworthy that BC-1215 inhibits FBXO3 from binding and degrading substrates but does not influence the protein level of FBXO3. Considering that HIPK2 has been proven to inhibit inflammation via Nrf2 in hypoxia/reoxygenation injury [27], it is tempting to speculate that FBXO3 may degrade HIPK2 by UPS to affect inflammation, leading to worse outcomes in I/R injury.

Taken together, the present study identifies an increase of FBXO3 expression in neurons and a peak at 24 h of reperfusion after stroke. We also demonstrate that FBXO3 exacerbates inflammation, leading to severe neuronal damage in I/R injury through the modulation of HIPK2. Our data suggest the therapeutic relevance of FBXO3 to ischemic stroke and provide a new perspective on the mechanism of I/R injury. However, the binding sites between FBXO3 and HIPK2 in ischemic stroke have not been clarified within the present study, the exact mechanism of FBXO3 on inflammation through ubiquitinating and degrading HIPK2 deserves further attention, and whether there are other crosstalk pathways of FBXO3 involved in inflammation will be the focus of our future research.

## 4. Materials and Methods

### 4.1. Animals

Adult male SD rats (60–80 days old, 180–220 g), purchased from Chongqing Medical University’s Animal Experimental Center, were bred under proper humidity, temperature, and light conditions, with food and water freely available. Rats were allocated randomly as follows: Sham group (*n* = 48, 0 died), MCAO 1 h/R 6 h group (*n* = 13, 1 died), MCAO 1 h/R 12 h group (*n* = 15, 3 died), MCAO 1 h/R 24 h group (*n* = 54, 6 died), MCAO 1 h/R 48 h group (*n* = 16, 4 died), MCAO 1 h/R 72 h group (*n* = 18, 6 died), NC group (*n* = 42, 6 died), si-FBXO3 group (*n* = 40, 4 died). All animal experimental protocols were approved by the Ethics Committee of Chongqing Medical University (Chongqing, China). All animal experiments were in strict accordance with the UK’s Animals (Scientific Procedures) Act 1986 and associated guidelines.

### 4.2. Cell Lines

Mouse hippocampal neuronal cell line HT22 cells, acquired from the Institute of Neuroscience of Chongqing Medical University, were maintained in Dulbecco’s Modified Eagle’s Medium (DMEM, Gibco, Shanghai, China) supplemented with 10% fetal bovine serum (FBS, VivaCell, Shanghai, China) and 1% penicillin–streptomycin (Beyotime Biotechnology, Shanghai, China) at 37 °C in a humidified 5% CO_2_ atmosphere.

### 4.3. Middle Cerebral Artery Occlusion/Reperfusion (MCAO/R) Model

The MCAO/R model was carried out according to the procedure we described previously [45]. After depriving male adult SD rats of food and drink for 8 h, their left middle cerebral artery (MCA) was occluded for 60 min with a nylon filament (Beijing Cinontech Co., Ltd., Beijing, China) under anesthesia (1% pentobarbital sodium, 45 mg·kg^−1^, intraperitoneal injection). To maintain the body temperature, heating pads were used under their bodies throughout the procedure. A 60 min occlusion later, we removed the filament to allow reperfusion. The reperfusion time accounted for 6, 12, 24, 48, and 72 h to meet the experimental demands. After MCAO/R, neurological deficit scores were conducted to assess the state of the model rats. The same procedure was operated with sham-treated animals, other than the occlusion of the MCA. To determine the endogenous expression changes of FBXO3 over different reperfusion times, we divided animals into six groups at random: (1) Sham group, (2) MCAO 1 h/R 6 h group, (3) MCAO 1 h/R 12 h group, (4) MCAO 1 h/R 24 h group, (5) MCAO 1 h/R 48 h group, and (6) MCAO 1 h/R 72 h group.

### 4.4. FBXO3 Interference in Rats

The design and synthesis of FBXO3 siRNA was accomplished by Sangon Biotech (Shanghai, China) We selected si-1211 (forward: 5′-CCACGAUUCCAUAUGGCAUTT-3′ and reverse: 5′-AUGCCAUAUGGAAUCGUGGTT-3′) for the best effect of FBXO3 knockdown (Appendix A). The negative control siRNA sequences were forward: 5′-UUCUCCGAACGUGUCACGUTT-3′ and reverse: 5′-ACGUGACACGUUCGGAGAATT-3′. FBXO3 siRNAs were injected into the left lateral cerebral ventricle of SD rats 24 h before MCAO modeling, with a dose of 1.5 OD per rat. The procedure of siRNA interference was described in our previous study [46].

The siRNA-transfected rats were divided into four groups at random: (1) Sham group, (2) MCAO group, (3) NC group (rats transfected with negative control siRNA then treated with MCAO/R), and (4) si-FBXO3 group (rats transfected with FBXO3 siRNA then treated with MCAO/R).

### 4.5. Neurological Function Tests

An improved scoring system was applied to assess the neurological function of rats after MCAO/R [47]. The scoring assay contained five grades: 0 means no neurological damage; 1 means incapable of extending the right forelimb fully; 2 means circling to the right; 3 means falling to the right side; 4 means losing consciousness or no signature of spontaneous autonomic activity. Observers were blinded to the group allocation of rats when scoring.

### 4.6. Quantification of Infarct Volume

Animals were sacrificed after reperfusion for 24 h and their brains removed immediately and cut evenly into five coronal sections (2 mm thick) prior to immersion in 5% TTC (Sigma-Aldrich, Shanghai, China) at 37 °C for 30 min. Then, the sections were placed in 4% paraformaldehyde at 4 °C overnight for fixation. Every stained section was photographed, and we quantified the infarct volume using ImageJ (version 6.0, NIH, Bethesda, MD, USA). The infarct volume was measured as follows: total lesion volume/total brain volume × 100%.

### 4.7. HE and Nissl Staining

After reperfusion for 24 h, rats were anesthetized, then injected with 4% paraformaldehyde at the apex to perfuse the brain. Afterward, the brains were removed and immersed in paraformaldehyde for more than a day, then dehydrated, embedded, and sliced into a thickness of 3 μm. Fixed sections were stained following standard protocols of hematoxylin and eosin (HE, Beyotime Biotechnology, Shanghai, China) staining or 0.1% cresol violet (Nissl, Beyotime Biotechnology, Shanghai, China). Both HE and Nissl sections underwent the same dewaxing, including soaking twice in xylene for 10 min and dehydrating with ethanol in different concentrations of 100% (5 min), 90% (2 min), and 70% (2 min) in sequence. After dehydration, sections for HE staining were immersed within hematoxylin staining solution for 10 min, 1% hydrochloric acid alcohol for 30 s, 1% ammonia for 30 s, and eosin for 5 min, with a 10 min rinsing executed between each step. As for Nissl staining, after soaking within Nissl staining solution for 10 min, sections were washed twice for 30 s each. Following dehydration, transparency, and mounting, all sections were observed using a light microscope.

### 4.8. OGD/R Treatment

The complete medium in HT22 cells was replaced with 01-057-1A DMEM free of glucose, glutamine, and sodium pyruvate (Biological Industries, Israel), and exposed to an incubator with 94% N_2_, 5% CO_2_, and 1% O_2_ at 37 °C for 4 h. Then, the cells were changed to normal culture conditions (5% CO_2_ and 95% air, at 37 °C, DMEM/high glucose supplemented with 10% of FBS and 1% penicillin–streptomycin) for 3, 6, 12, 24, and 48 h Subject to experimental requirements. Six groups were used to observe the expression of FBXO3 over different reoxygenation times: (1) Control group, (2) OGD 4 h/R 3 h group, (3) OGD 4 h/R 6 h group, (4) OGD 4 h/R 12 h group, (5) OGD 4 h/R 24 h group, and (6) OGD 4 h/R 48 h group.

### 4.9. FBXO3 siRNA and FBXO3 Inhibitor BC-1215 Interference in Cells

The FBXO3 siRNA (si-480, as in Appendix A) was designed and synthesized as follows: forward: 5′-CGGAUGAUUAUCGCUGUUCAUTT-3′ and reverse: 5′-AUGAACAGCGAUAAUCAUCCGTT-3′. The negative control siRNA sequences were forward: 5′-UUCUCCGAACGUGUCACGUTT-3′ and reverse: 5′-ACGUGACACGUUCGGAGAATT-3′. One day before OGD/R (the reoxygenation time was 24 h), FBXO3 siRNA or negative control siRNA was added into media along with the transfection reagent, Lipofectamine 2000 (Invitrogen; Thermo Fisher Scientific, Inc., Shanghai, China). The study was assigned into four groups: (1) Control group, (2) OGD group, (3) NC group (cells transfected with negative control siRNA then treated with OGD/R), and (4) si-FBXO3 group (cells transfected with FBXO3 siRNA then treated with OGD/R).

BC-1215 (MedChemExpress, Shanghai, China) was dissolved to 10 mM/mL with DMSO. HT22 cells were exposed to BC-1215 (10 μg/mL) or DMSO (10 μg/mL) for 2 h, followed by OGD/R treatment. Similarly, four groups were set up: (1) Control group, (2) OGD group, (3) DMSO group (cells treated with DMSO then treated with OGD/R), and (4) BC-1215 group (cells treated with BC-1215 then treated with OGD/R).

### 4.10. Cell Viability

The CCK-8 assay kit (Topscience, Shanghai, China) was applied to assess neuronal cell viability. HT22 cells were cultured in 96-well plates and subjected to siRNA treatment, followed by OGD/R. Then, the cells were incubated with CCK-8 solution at a concentration of 10 μL/mL at 37 °C for at least 30 min. Plates were measured with a microplate reader at an optical density of 450 nm.

### 4.11. Flowcytometry

HT22 cells were treated with FBXO3 siRNA along with OGD/R treatment, followed by three washes by PBS. After incubating with Annexin V-FITC and Propidium Iodide (PI), the fluorescent intensity was analyzed by flow cytometry (FCM) (FACS Vantage SE, Becton Dickinson, San Jose, CA, USA), and FlowJo (BD Biosciences, Wokingham, UK) was used for analysis.

### 4.12. Calcein/PI Cell Viability/Cytotoxicity Assay

In order to survey living and dead cells simultaneously under a microscope, HT22 cells were cultured using glass-bottomed cell culture dishes. After si-FBXO3 and OGD/R treatment, cells were incubated with a Calcein/PI staining working solution (Beyotime Biotechnology, Shanghai, China) for 30 min. Images were captured on a LeicaTCS SP8 confocal laser scanning microscope (Wetzlar, Germany).

### 4.13. Western Blotting

The ischemic brain tissues and HT22 cells were homogenized with a cell lysis buffer for Western blotting and IP (Beyotime Biotechnology, Shanghai, China) comprising 1% phenylmethanesulfonyl fluoride (PMSF, Beyotime Biotechnology, Shanghai, China). The loading buffer was mixed with supernatants. Different percentages of SDS-PAGE gels were chosen for separating 50 μg of total protein on the basis of different protein molecular weights. Then, the wanted proteins were electrotransferred to polyvinylidene fluoride (PVDF) membranes at different times based on protein molecular weights. After blocking, the protein bands were incubated with relative primary antibodies including FBXO3 (1:400, Santa Cruz Biotechnology, Texas, USA, sc-514625), β-actin (1:1000, Proteintech, Wuhan, Hubei, China, 66009-1-Ig), IL-1β (1:1000, ABclonal, A1112), IL-18 (1:1000, Proteintech, 10663-1-AP), TNF-α (1:1000, ABclonal, Wuhan, Hubei, China, A11534), and HIPK2 (1:400, Santa Cruz Biotechnology, sc-100383) overnight at 4 °C. Twelve hours later, secondary antibodies were used to incubate the membranes for 1 h. An imaging densitometer (Bio-Rad, Hercules, CA, USA) was applied to detect the densities of the bands, whose gray values were quantified by ImageJ.

### 4.14. Co-Immunoprecipitation (Co-IP) Assay

Co-IP operations followed the Beaver Beads™ Protein A (or A/G) Immunoprecipitation Kit’s instruction (Beaver Nano-Technologies, Suzhou, China). Briefly, an FBXO3 antibody (1:50, Santa Cruz Biotechnology, sc-514625) was incubated with magnetic beads for 60 min to ensure stable binding between the antibody and beads, with mouse IgG (1:50, ABclonal, AC011) as the negative control. Then, the complex was added to HT22 cell samples, incubating them on a rotary mixer at 4 °C overnight. Afterward, the beads–antibody–antigen complex was blended with loading buffer preceding centrifugation at a speed of 13,000× *g* for 10 min. Lastly, the extracted supernatants were denatured at 95 °C for 5 min for subsequent immunoblotting, as described above.

### 4.15. Quantitative PCR (qPCR) Assay

Total RNA from ischemic penumbra tissue of cortex or HT22 cells was extracted using TRIzol Reagent (Invitrogen, Carlsbad, CA, USA). PrimeScript™ RT reagent Kit with gDNA Eraser (Perfect Real Time) (Takara Biomedical Technology, Beijing, China) was used for the reverse transcription of RNA for obtaining cDNA. The real-time evaluation of PCR was conducted with CFX manager 3.0 software (Bio-Rad, Hercules, CA, USA) using SYBR Green Master Mix (Thermo Fisher Scientific, Shanghai, China). The glyceraldehyde 3-phosphate dehydrogenase (GAPDH) expression was set as an endogenous reference gene. The oligonucleotide sequences involved are listed in Table 1.

### 4.16. Enzyme-Linked Immunosorbent Assay (ELISA)

All ELISA experiments followed the manufacturers’ instructions, and the levels of IL-1β, IL-18, and TNF-α were measured by IL-1β, IL-18, and TNF-α ELISA kits (Quanzhou jiubang Biotechnology, Quanzhou, Fujian, China) using brain tissue lysates and supernatants of HT22 cells. Briefly, each antigen-coated well was added with standards in different concentrations and samples separately, then mixed them with 100 μL HRP conjugate reagent, followed by sealing and 1 h incubation at 37 °C. After removing the liquid from the wells, washing was performed five times with a washing solution. Finally, solutions A and B were added to the plates for 15 min free from light, and we subsequently added a stop solution. The detection was at the absorbance of 450 nm.

### 4.17. Immunofluorescence (IF) Analysis

Brain sections (3 μm thickness) were dewaxed and placed in a box with an EDTA antigen repair buffer (PH 8.0), then put in a microwave to repair the antigen. Meanwhile, the coverslips of HT22 cells were fixed, permeabilized, and blocked. The primary antibodies, including NeuN (1:50, ABclonal, A19086), Iba-1 (1:50, Abcam, Boston, MA, USA, ab178847), GFAP (1:50, ABclonal, A19058), FBXO3 (1:50, Santa Cruz Biotechnology, sc-514625; 1:50, CUSABIO, CSB-PA892132LA01HU), and HIPK2 (1:50, Santa Cruz Biotechnology, sc-100383), along with MPO (1:50, Servicebio, Wuhan, Hubei, China, GB11224) were used at 4 °C for 12 h. Subsequently, brain sections and coverslips were incubated with CoraLite488-conjugated Goat Anti-Mouse IgG antibody (1:200, Proteintech, SA00013-1) and CoraLite594-conjugated Goat Anti-Rabbit IgG antibody (1:200, Proteintech, SA00013-4) for 1 h away from light. Ten minutes after dropping Antifade Mounting Medium with DAPI (Beyotime Biotechnology, Shanghai, China) on the coverslips and sections, visualization was accomplished with a laser scanning confocal microscope (LSCM).

### 4.18. Statistical Analysis

All data are exhibited as means ± SD (standard errors), and images were produced by GraphPad Prism software (version 6.0) (San Diego, CA, USA). Comparisons between groups were conducted by Student’s *t*-test. A *p*-Value of <0.05 indicated a statistically significant difference.

## Figures and Tables

**Figure 1 ijms-23-13648-f001:**
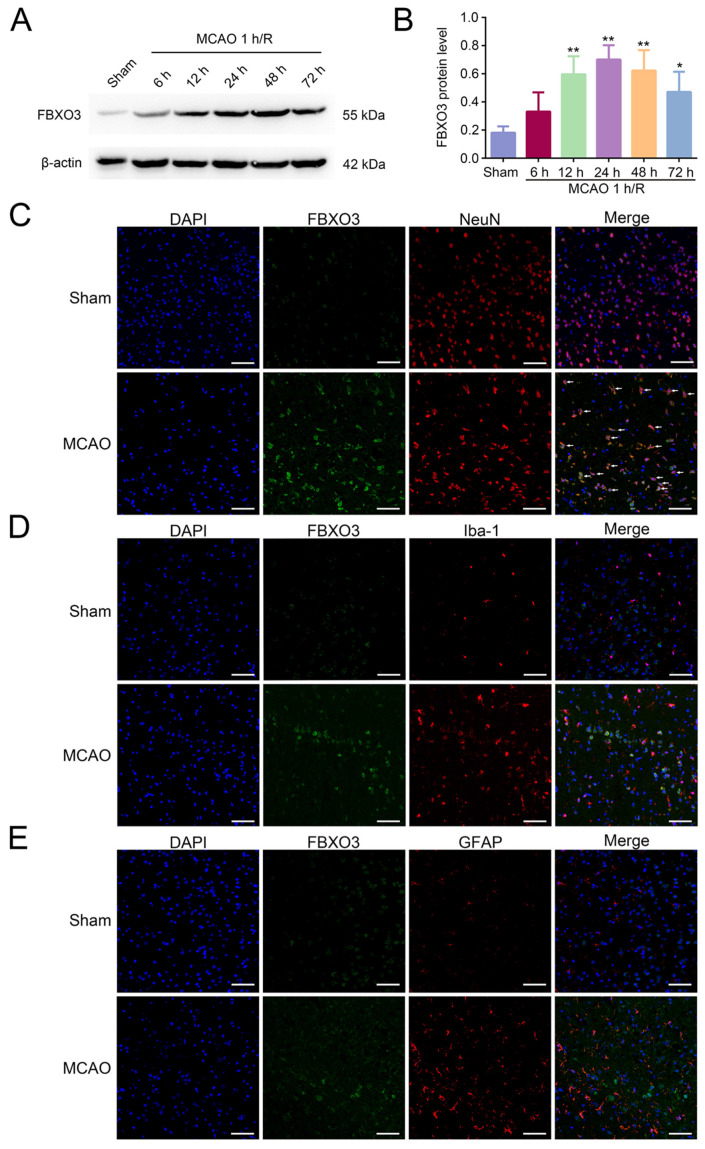
FBXO3 was elevated after Middle cerebral artery occlusion/reperfusion (MCAO/R) in Sprague-Dawley (SD) rats, and specifically expressed in neurons. (**A**,**B**) Western blotting (WB) analysis of FBXO3 in the peri-infarcted cortex of SD rats at MCAO 1 h/R 6, 12, 24, 48, 72 h, and sham treatment. (**C**–**E**) Representative images (400×, scale bar = 100 μm) of FBXO3 (green)/NeuN (neuronal biomarker, red)/DAPI (blue), FBXO3/Iba-1 (microglial biomarker, red)/DAPI, and FBXO3/GFAP (astrocytic biomarker, red)/ DAPI immunostaining in the ischemic penumbra at sham and MCAO 1 h/R 24 h treatment, respectively. The arrows in Figure 1C indicate representative FBXO3+ neurons. Statistics for each group are expressed as mean ± SD (*n* = 6). * *p* < 0.05, ** *p* < 0.01 (values in Sham group versus different reperfusion time group).

**Figure 2 ijms-23-13648-f002:**
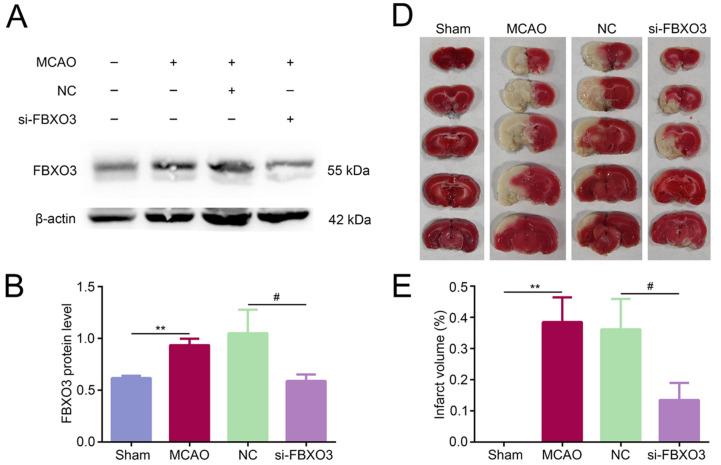
Interference of FBXO3 rescued the neurological outcomes after MCAO/R in vivo. (**A**–**C**) WB and qPCR analysis of FBXO3 in the peri-infarcted cortex of MCAO/R rats with or without siRNA treatment and in sham rats. (**D**) Representative TTC (2,3,5-triphenyltetrazolium chloride) staining images of coronal sections of MCAO/R rats with or without siRNA treatment and sham rats. (**E**) Quantification of infarct volumes of coronal sections of MCAO/R rats with or without siRNA treatment and sham rats by ImageJ (total lesion volume/total brain volume × 100%). (**F**) Neurological deficit scores of MCAO/R rats with or without siRNA treatment and sham rats. (**G**) HE and Nissl staining (400×, scale bar = 50 μm) in the peri-infarcted cortex of MCAO/R rats with or without siRNA treatment and in sham rats. The arrows indicate representative Nissl bodies. Statistics for each group are expressed as mean ± SD (*n* = 6). ** *p* < 0.01, **** *p* < 0.0001 (values in Sham group versus MCAO group), and # *p* < 0.05, #### *p* < 0.0001 (values in NC group versus si-FBXO3 group).

**Figure 3 ijms-23-13648-f003:**
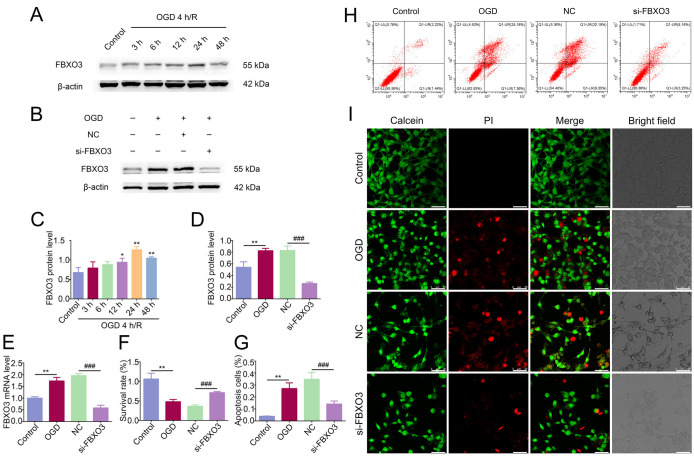
Interference of FBXO3 attenuated the glucose deprivation/reoxygenation (OGD/R) induced neuronal death in vitro. (**A**,**C**) WB analysis of FBXO3 in HT22 cells at OGD 4 h/R 3, 6, 12, 24, and 48 h. (**B**–**E**) WB and qPCR analysis of FBXO3 in HT22 cells at OGD 4 h/R 24 h with or without siRNA treatment. (**F**) CCK8 assays of HT22 cells at OGD 4 h/R 24 h with or without siRNA treatment. (**G**–**H**) Quantification and representative images of Flowcytometry by detecting the fluorescent intensity of Annexin V-FITC and Propidium Iodide (PI) of HT22 cells at OGD 4 h/R 24 h with or without siRNA treatment. (**I**) Representative immunofluorescence images (400×, scale bar = 50 μm) of Calcein and PI in HT22 cells at OGD 4 h/R 24 h with or without siRNA treatment. Statistics for each group are expressed as mean ± SD (*n* ≥ 6). * *p* < 0.05, ** *p* < 0.01 (values in Control group versus different reoxygenation time group), ### *p* < 0.001 (values in NC group versus si-FBXO3 group).

**Figure 4 ijms-23-13648-f004:**
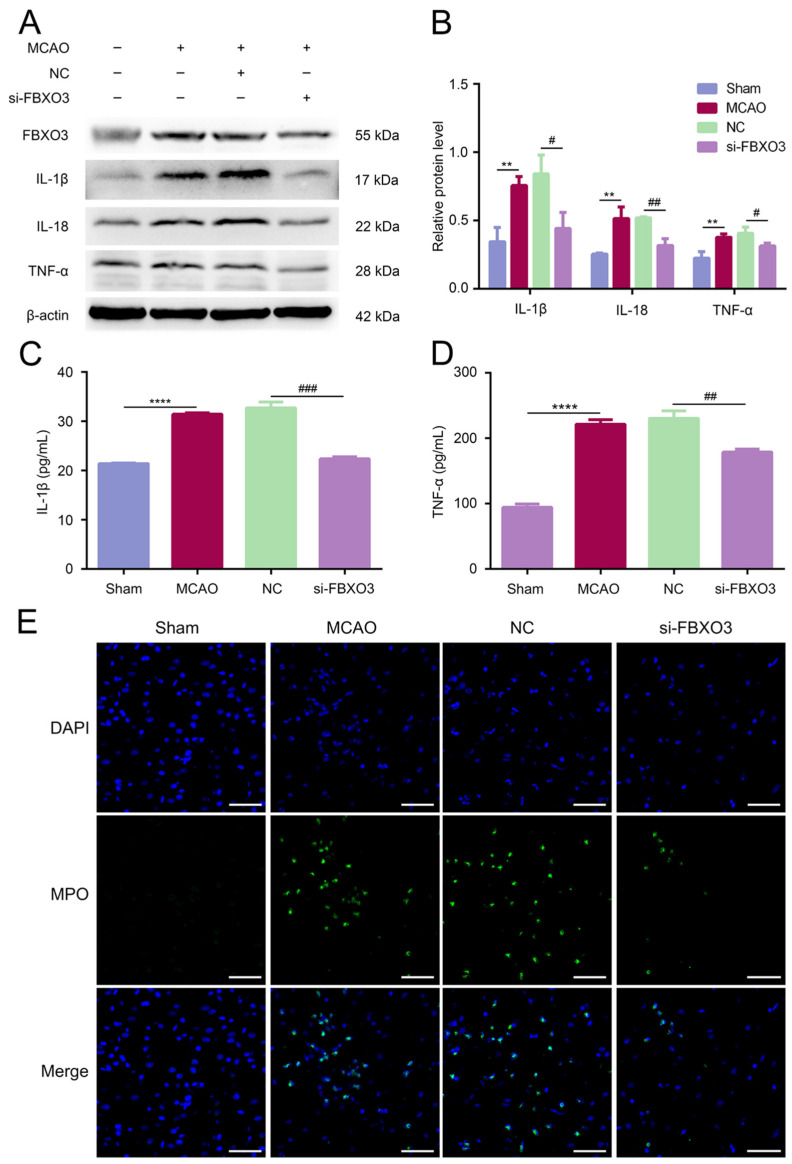
Treatment of si-FBXO3 inhibited inflammatory response induced by ischemia/reperfusion (I/R) injury in vivo. (**A**,**B**) WB analysis of inflammatory cytokines including IL-1β, IL-18, and TNFα in the peri-infarcted cortex of MCAO/R rats with or without siRNA treatment and in sham rats. (**C**,**D**) ELISA analysis of inflammatory cytokines including IL-1β and TNFα in the peri-infarcted cortex of MCAO/R rats with or without siRNA treatment and in sham rats. (**E**) Immunostaining of MPO (inflammation indicator, green) and DAPI (blue) in the peri-infarcted cortex of MCAO/R rats with or without siRNA treatment and in sham rats. (×630, scale bar = 100 μm). Statistics for each group are expressed as mean ± SD (*n* ≥ 6). ** *p* < 0.01, **** *p* < 0.0001 (values in Control group versus MCAO group), # *p* < 0.05, ## *p* < 0.01, ### *p* < 0.001 (values in NC group versus si-FBXO3 group).

**Figure 5 ijms-23-13648-f005:**
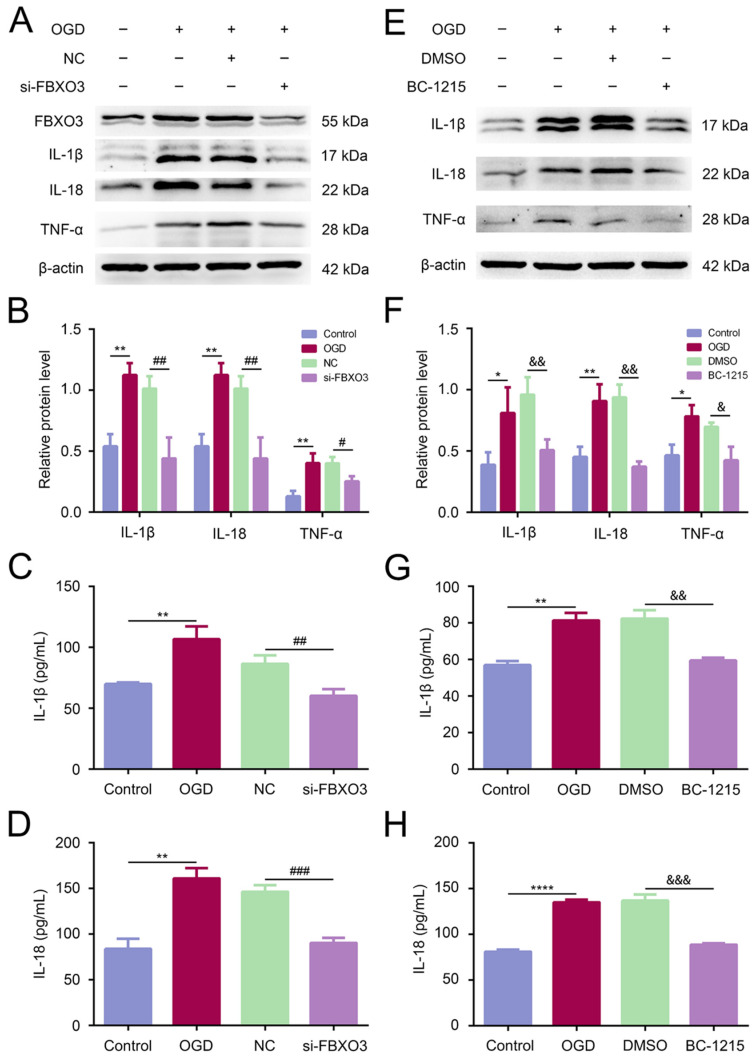
Inhibition of FBXO3 alleviated inflammatory response induced by I/R injury in vitro. (**A**,**B**) WB analysis of inflammatory cytokines including IL-1β, IL-18, and TNFα in HT22 cells at OGD 4 h/R 24 h with or without siRNA treatment. (**C**,**D**) ELISA analysis of inflammatory cytokines including IL-1β and IL-18 in HT22 cells at OGD 4 h/R 24 h with or without siRNA treatment. (**E**,**F**) WB analysis of inflammatory cytokines including IL-1β, IL-18, and TNFα in HT22 cells at OGD 4 h/R 24 h with DMSO or BC-1215 treatment. (**G**,**H**) ELISA analysis of inflammatory cytokines including IL-1β and IL-18 in HT22 cells at OGD 4 h/R 24 h with DMSO or BC-1215 treatment. Statistics for each group are expressed as mean ± SD (*n* ≥ 6). * *p* < 0.05, ** *p* < 0.01, **** *p* < 0.0001 (values in Control group versus MCAO group), # *p* < 0.05, ## *p* < 0.01, ### *p* < 0.001 (values in NC group versus si-FBXO3 group), & *p* < 0.05, && *p* < 0.01, &&& *p* < 0.001 (values in DMSO group versus BC-1215 group).

**Figure 6 ijms-23-13648-f006:**
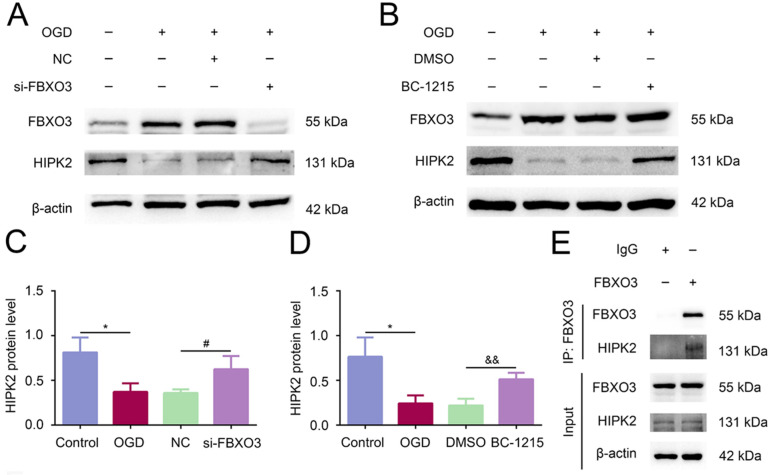
FBXO3 facilitated inflammation probably through binding and degrading HIPK2 in HT22 cells after OGD/R stimulation. (**A**,**C**) WB analysis of HIPK2 in HT22 cells at OGD4 h/R 24 h with or without siRNA treatment. (**B**,**D**) WB analysis of HIPK2 in HT22 cells at OGD4 h/R 24 h with DMSO or BC-1215 treatment. (**E**) Co-Immunoprecipitation (Co-IP) analysis of FBXO3 and HIPK2 in HT22 cells. (**F**) Representative immunofluorescence images (200×, scale bar = 100 μm) of FBXO3 (red) and HIPK2 (green) in HT22 cells at OGD4 h/R 24 h with or without siRNA treatment. Statistics for each group are expressed as mean ± SD (*n* ≥ 6). * *p* < 0.05 (values in Control group versus MCAO group), # *p* < 0.05 (values in NC group versus si-FBXO3 group), && *p* < 0.01 (values in DMSO group versus BC-1215 group).

**Figure 7 ijms-23-13648-f007:**
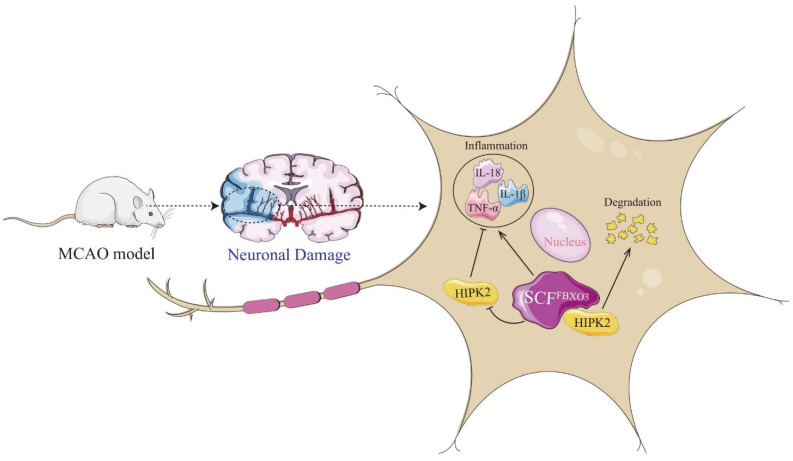
Schematic illustration of FBXO3 driving neuroinflammation to aggravate neuronal damage in cerebral I/R injury by degrading HIPK2.

**Table 1 ijms-23-13648-t001:** Primer sequences for qPCR.

Gene Name	Primer Sequences
GAPDH (rat)	Forward: AGTTCAACGGCACAGTCAAG Reverse: TACTCAGCACCAGCATCACC
FBXO3 (rat)	Forward: GGACCTGGAGTAGTTGGTGAA Reverse: CATGTCTGCTGAGTCATCGT
GAPDH (mouse)	Forward: GGTTGTCTCCTGCGACTTCA Reverse: TGGTCCAGGGTTTCTTACTCC
FBXO3 (mouse)	Forward: GGTGTCTATAGCTCGATTGGAA Reverse: TCATCTGACTCATTCTCATCCG

## Data Availability

Not applicable.

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
