# Peer review of "E3 Ubiquitin Ligase FBXO3 Drives Neuroinflammation to Aggravate Cerebral Ischemia/Reperfusion Injury"

_ijms, 2022, doi:10.3390/ijms232113648_

Round 1

Reviewer 1 Report

The manuscript presents results of the authors' experimental research of the mechanisms underlying the ischemia/reperfusion (I/R)  injury in ischemic stroke. It is a "hot" topic, since the emergency IV or endovascular infusion of recombinant TPA is quite common and so is its complication - the I/R  injury. The used animal model adequately imitates the clinical situation. The authors got interesting results which will attract attention of other researchers involved in the I/R  injury research and stroke mechanisms research in general.

Unfortunately, the manuscript needs extensive English language editing. Some sentences are just difficult to understand, for example this sentence taken from the Abstract: "Furthermore, FBXO3 was verified to dysregulate the protein level of Homeodomain Interacting Protein Kinase 2 (HIPK2) likely through ubiquitin-proteasome system (UPS) to expedite cerebral I/R injury by administration of FBXO3 siRNA and BC-1215". Because of poor English, it is difficult to read especially the Results section. Besides, the authors should specify which exactly neurological deficit score they were using. 

Reviewer 2 Report

I have reviewed the paper with great interest since E3 ligases play vital role in UPS system by knockout degradation of target proteins by PROTACS method that can target undruggable. The major goal of the study is to investigate whether FBXO3 exerts impacts on cerebral I/R injury. The authors have employed expression analysis and cellular assays to evaluate the pro-inflammatory effect of FBXO3 by ubiquitylating and degrading HIPK2. The paper is generally well written, methods are well described and results are well discussed. The data presented in the paper supports the conclusion of the paper. Although I feel the work is not ground breaking, this is an important addition to the literature. I recommend this paper for publication after addressing couple of my minor comments.

1.      The authors may add a statement about the limitation and future prospect of the current study in the abstract.

2.      The authors must add some more details about the molecules studied in the introduction section by citing relevant literature (For example https://doi.org/10.4155/fmc-2021-0157,https://doi.org/10.1074/jbc.M116.771667)

Round 2

Reviewer 1 Report

Good luck!